# Correlation between Buccal Bone Thickness at Implant Placement in Healed Sites and Buccal Soft Tissue Maturation Pattern: A Prospective Three-Year Study

**DOI:** 10.3390/ma13030511

**Published:** 2020-01-21

**Authors:** Davide Farronato, Pietro Mario Pasini, Andrea Alain Orsina, Mattia Manfredini, Lorenzo Azzi, Marco Farronato

**Affiliations:** 1School of Dentistry, Department of Medicine and Surgery, University of Insubria, 21100 Varese, Italy; davide@farronato.it (D.F.); pm.pasini@gmail.com (P.M.P.); aa.orsina@gmail.com (A.A.O.); 2Corso Europa 10, 20122 Milan, Italy; 3Department of Medicine and Surgery, University of Insubria, Unit of Oral Pathology, Dental Clinic, ASST dei Sette Laghi, 21100 Varese, Italy; lorenzoazzi86@hotmail.com; 4IRCCS Fondazione Cà Granda, University of Milan, Via Francesco Sforza 28, 20122 Milan, Italy; marcofarronato@msn.com

**Keywords:** dental implant, buccal bone, soft tissue, recession

## Abstract

Background: Optimal aesthetic implant restoration is a combination of a visually pleasing prosthesis and adequate surrounding peri-implant soft tissue architecture. This study describes the influence of the residual buccal bone thickness (BBT), measured at the time of implant placement, on the soft tissue maturation during three years of follow-up. Methods: Seventy-eight implants were enrolled in the present study. The BBT was assessed at the surgical stage and each case assigned to Group 1 (BBT values ≤0.5 mm), Group 2 (BBT values >0.5 and <1.5 mm), or Group 3 (BBT values ≥1.5 mm). Only native bone and healed sites were included. The tooth height (TH), based on the distance between the buccal free gingival margin at the zenith level and the crown incisal edge, according to the main axis of the tooth, was monitored at one, two, and three years from the final prosthodontic rehabilitation to determine any occurrence of recession or coronal repositioning of the gums over time. A Pearson Two-Tailed test was applied and the significance level set at *p* ≤ 0.05. Results: For BBT values ≤0.5 mm, the buccal gum at three years showed an average recession of 1.22 ± 0.41 mm. For BBT values >0.5 and <1.5 mm, the buccal gum also showed recession of 0.64 ± 0.29 mm. In contrast, for BBT values ≥1.5 mm, the buccal gum showed coronal growth of 0.77 ± 0.22 mm. The differences between the groups were significant (*p* ≤ 0.01) at all times. Conclusion: The BBT at the time of implant placement was found to affect the buccal gingival margin stability over three years of observation.

## 1. Introduction

The aesthetic of the prosthesis and tissues has been identified as closely linked to the satisfaction of both patients and clinicians [1]. While implant rehabilitation in the aesthetic zone is a common treatment, the aesthetic and functional outcomes and variables affecting peri-implant tissue maturation still remain important topics of research [2]. Accordingly, what are the variables that can influence the aesthetic outcome of implant rehabilitation? One of the main factors is the tridimensional position of the fixture, as described in previous literature [3,4,5], along with the quality and quantity of both hard and soft tissue [2]. Studies by Linkevicius et al. have focused on the importance of a thick or thin biotype to improve immediate implant success and prevent future mucosal recession [6], as soft tissue will always adapt to the condition of the local setting [7,8]. Meanwhile, Steigman et al. defined the importance of the emergence profile to manage soft tissue adjacent to dental implants [9]. Thus, soft tissue management at the prosthetic and surgical stages is considered essential to gain predictable and aesthetic results [10,11]. Moreover, the implant and abutment design, along with the connection, are also known to play important roles [12,13]. As regards to the quantity of hard tissue, Buser et al. investigated the ideal amount of hard tissue required around dental implants [14] and found that the thickness of the buccal bone at the implant site played a fundamental role in the aesthetic predictability of the rehabilitation [14,15]. Plus, Yoda et al. suggested a minimum initial buccal bone thickness of 1.5 m [16]. Although still a topic of discussion, the aesthetic outcome of an implant-supported restoration would seem to be undoubtedly related to the buccal soft tissue height [10,17]. Therefore, this study focuses on the role of the buccal bone thickness (BBT) on facial tissue stability, measured at the time of implant placement. To reduce the confounding variables, the protocol focused on healed sites only. The same implant and prosthetic protocol was used for all the examined samples. The ideal thickness of the buccal bone at implant placement was then investigated in order to minimize the risk of tissue recession during biological maturation.

## 2. Materials and Methods

### 2.1. Study Design

This study was designed as a prospective observation of consecutively enrolled patients with strict exclusion criteria, where the patients receive the same type of implant at healed edentulous sites and were rehabilitated using the prosthetic and surgical protocols described below. All the surgical and prosthetic procedures were performed by the same clinician.

### 2.2. Patients Selection

This study was conducted in accordance with the fundamental principles of the Helsinki Declaration, and approved by the Ethics Committee of the University of Insubria (N° 826 for protocol 0034086). Between October 2014 and March 2016, a total of 45 consecutive patients, referred to the authors’ dental clinic for implant rehabilitation of partially edentulous ridges, were identified as candidates for dental implant treatment based on clinical and radiographic examination. The inclusion criteria included single or multiple mandibular or maxillary partial edentulism, the need for implant rehabilitation, and age ≥18 at the time of surgery. The exclusion criteria included a full-mouth plaque score (FMPS) ≥25%, full-mouth bleeding score (FMBS) ≥25%, abuse of alcohol or drugs, the presence of acute oral infections, drugs correlated to any gum hyperplastic response, remote or recent radiation therapy in the oromaxillofacial area, recent chemotherapy, uncontrolled diabetes, pregnancy, smoking habit (>10 cigarettes/day), and any other systemic disease or syndrome that may affect gum and bone behavior. Additional exclusion criteria included a lack of facial keratinized mucosa (clinically measured using a periodontal probe, UNC15, University of North Carolina, USA), need for soft tissue or bone augmentation at any stage of the treatment, implants that showed any sign of evident inflammation of the peri-implant mucosa over time [18], and implants that were not osseointegrated after the healing period or after being restored with a fixed prosthesis. Plus, patients with a lack of collected data, at any stage of the planned follow-up, were also excluded from this prospective study. If an implant could not be placed 0.5 mm below the bone crest, this piece of data was immediately excluded from the study and not considered in the statistical analysis. Similarly, data were excluded when the fixture could not be placed following the surgical protocol, mainly due to the presence of an interproximal steep bone peak, as the implant could not be placed in native bone alone. As a result, a total of 98 implants were placed according to standardized surgical and prosthetic protocols (described below), which were pre-established at the beginning of the study, and 78 implants were finally considered in the statistical analysis (7 patients were lost from the study, representing 20 implants, where 17 implants had no follow-up data, and 3 implants experienced mucositis occurred during the observation period.

### 2.3. Implants

All the implants used in this study were root-form conical implants with a knife-edge thread design (MegaGen AnyRidge, MegaGen Implant Co. Ltd., Daegu, Korea). The implant connection was a five-degree per side conical connection with a switching platform. The implant was fabricated of Cp Ti, considered a reliable and noncytotoxic material [19], while the surface treatment was an SLA technique with an incorporated nano-layer of Ca^2+^, which has been proven to have an elevated cytocompatibility [20].

### 2.4. Surgical Protocol

The patients were clinically (Figure 1) and radiographically evaluated to determine the correct treatment plan. Panoramic radiography was used as the first-level exam. CBCT (Cone beam computed tomography) was also performed to assess the native bone volume. Oral hygiene, soft tissue health, amount of keratinized gingiva, stability of the remaining teeth, and other factors potentially influencing the treatment plan were all clinically evaluated. Antibiotic prophylaxis, consisting of amoxicillin + clavulanic acid 2 g/day for 6 days, or clindamycin 600 mg/day for 6 days in the case of penicillin-allergic patients, was prescribed to reduce the risk of infections, starting with 2 g one hour before surgery. Anti-inflammatory therapy with NSAIDs, consisting of Ketoprofen Lysine 80 mg/day for 3 days, starting with 80 mg 1 h before surgery, was also recommended. The local anesthesia was articaine + adrenalin 1:50,000 at the site of intervention and articaine + adrenalin 1:100,000 at other sites.

A two-stage surgical technique was used for the implant placements. For the first stage surgery, a full-thickness buccal flap was carefully sculpted with help of an elevator, and the vertical thickness of the soft tissue was measured using a 1.0 mm marked periodontal probe (UNC15, University of North Carolina, USA). If the vertical soft tissue thickness was 2 mm or less, the tissue was considered thin and the patient was excluded from the present study [21]. Thereafter, the lingual full-thickness flap was elevated and the implant placement site prepared. The full-thickness mucoperiosteal flap (Figure 2) was elevated to give access to the underlying alveolar bone, which allowed assessment of the native bone present and correct tridimensional positioning of the fixture.

The implant sites (Figure 3) were prepared at 800 rpm with abundant irrigation to avoid any thermal osteonecrosis [22], according to the manufacturer’s instructions.

When positioning two or more implants, a direction pin was used in the first implant site to ensure parallel insertion of the other implants. The inter implant distance was at least 3 mm [23]. All the fixtures were inserted using a torque-controlled driver at 60 rpm (Figure 4) [24] and finally positioned 0.5 mm subcrestal to the crestal edge of the buccal bone, according to the manufacturer’s instructions (Figure 5).

A torque value between 30 and 50 Nm was deemed acceptable, depending on the bone density. The mucoperiosteal flap was reconnected using a tension-free suture (Figure 6) to facilitate wound healing and avoid bacterial contamination. All the fixtures were placed in native crestal bone, and no implant was placed in a post-extractive or regenerated site. After implant placement, all the patients observed a 10-day liquid diet, followed by another 15 days of soft foods. The post-surgery oral hygiene protocol included a modified Bass brushing technique and 0.2% chlorhexidine rinsing three times a day for one week. All the patients were advised against using removable partial dentures with mucosal support over the surgery site. In some cases, an adhesive provisional Maryland bridge, that avoided any contact with the surgical site was applied.

### 2.5. Prosthetic Protocol

The osteointegration time was 3 months for the mandibular fixtures and 6 months for the maxillary fixtures. For the second-stage surgery, a midcrestal mesiodistal flap was sculpted to enhance the quantity of keratinized tissue. The cover screw was manually removed using a dedicated screwdriver and replaced with a healing abutment. Alginate impressions were taken to create individualized impression trays, which were then used to take secondary impressions using a polyether and pick-up technique. Two weeks after the second-stage surgery, a resin screw-retained provisional prosthesis was delivered. After 3 months, the provisional restoration was removed and another impression taken to manufacture the definitive metal-ceramic prosthesis. All the rehabilitations were cemented on milled abutments, where the abutment diameter at the connection was smaller than the implant platform diameter, according to the platform switching concept. The crown finishing line was never deeper than 1 mm from the free gingival margin to allow for adequate cement removal, plus a periapical X-ray was used to check for any excess radiopaque cement [25,26]. The occlusion was checked to remove any pre-contacts or interferences in centric, lateral, or protrusive movements. The definitive rehabilitations included both single crowns and partial fixed prostheses to a maximum of 4 elements.

### 2.6. Data Collection

Using patient information from clinical records and anamnesis, the following patient variables were analyzed: sex, age, smoking habit and number of cigarettes/day, systemic diseases, chronic or aggressive periodontitis, parafunctions (bruxism, clenching), level of oral hygiene [27], and number of professional oral hygiene procedures in the last 3 years. The implant variables at the time of the first surgery included the length, diameter, jaw position (maxilla/mandible/anterior/posterior), torque, and BBT at the time of implant placement within 0.5 mm of the crest edge. Clinical and radiological (intra-oral radiograph) evaluations of each implant were conducted at the time of implant placement, provisional placement, final crown delivery, and 1-, 2-, and 3-year follow-up visits after the final rehabilitation. Before any radiological examination, a film holder was directly customized in the patient’s mouth using auto-polymerizing acrylic resin to reproduce the same position of the film, thereby obtaining superimposable dental radiographs at different time intervals. The implant placement depth was measured during the positioning using a handpiece connector with millimeter marks and then checked using a periapical radiograph after the implant placement. The average distance from the implant shoulder to the first radiographic bone peak was also recorded. Any implant that was positioned within 0.5 mm of the crest edge in the periapical radiograph was not considered in the study. Buccal bone thickness (BBT) measurements were taken intraoperatively at the time of implant placement using a sterile stainless steel dental gauge caliper with a 1/10 mm resolution (Wise Instruments, Valley Stream, NY, USA). This tool allowed the BBT to be measured 0.5 mm below the coronal peak of the bone above the implant shoulder (Figure 7 and Figure 8a,b).

All the measurements were taken by the same clinician who performed both the surgical and the prosthetic procedures, and the mean was used in the statistical analysis based on a 95% confidence interval (CI). At the time of the final crown delivery, the tooth height (TH) was measured from the buccal free gingival margin at the zenith level to the crown incisal edge, according to the main axis of the tooth (Figure 9a,b). This measurement was performed using a periodontal probe (UNC15, University of North Carolina, Chapel Hill, NC, USA) and repeated at the 1-, 2-, and 3-year follow-up visits from the delivery of the final crown. The baseline was set as the time of delivering the final prosthesis. The presence of keratinized facial mucosa was also measured.

### 2.7. Data Recording and Statistical Analysis

The statistical analysis included 78 consecutively enrolled implants, where 46 were placed in the lower jaw (L) and 32 in the upper jaw (U). The total follow-up time was 3 years from the final prosthesis delivery, which occurred 6 to 8 months after the provisional crown positioning. The TH baseline was set as the time of placing the final crown, and the follow-up values were measured at 1-, 2-, and 3-year follow-up visits. All the data were analyzed using statistical software (SPSS 20, IBM). The implant values were divided into 3 BBT ranges: Group 0 ≤0.5 mm, Group 1 >0.5 to <1.5 mm, and Group 2 ≥1.5 mm. The TH measurements were performed using a probe (Periodontal probe UNC 15, University of North Carolina, USA). Delta mean values were used in the statistical analysis based on a 95% CI. The delta TH (ΔTH) means and statistical correlations (Pearson Two-Tailed 95% CI) were analyzed according to the BBT grouping and patient’s correlation to the analyzed values. The ΔTH was determined by subtracting the follow-up TH from the baseline TH value, where positive values represent recession and negative values represent coronal repositioning of the gum as regards to the final prosthodontic rehabilitation. In particular, recession occurs when the buccal free gingival margin at the zenith repositions, over time, apically to the implant, whereas growth is associated with coronal migration of the gum. A probability (*p*) value ≤ 0.05 was considered statistically significant, *p* ≤ 0.01 highly significant, and *p* > 0.05 not significant.

## 3. Results

Among the 78 implants considered in the statistical analysis, eight were placed in smokers (10.3%) and 70 in nonsmokers (89.7%), and 55 were placed in females (70.5%) and 23 in males (29.5%). Forty-six fixtures were placed in the mandible (58.9%) and 32 in the maxilla (41.1%). The average BBT was 0.96 ± 0.58 mm and the average recession at three years was 0.34 ± 1.75 mm. The BBT group distribution was as follows: Group 0 included 23 implants (29.5%), Group 1 included 29 (37.2%), and Group 2 included 26 (33.3%). In the statistical analysis, no correlation was found at three years between the ΔTH and sex (*p* = 0.7), smoking habits (*p* = 0.27), or upper or lower jaw positioning (*p* = 0.26).

However, a consistent correlation was found between the BBT and the ΔTH (*p* ≤ 0.001) (Table 1) and the BBT grouping and the ΔTH (*p* ≤ 0.001) over the three-year follow-up (Table 1). Therefore, the results indicate that recession or growth of the gingival buccal margin is affected by the thickness of the buccal bone at the time of surgery.

The average recession (ΔTH) at three years was 1.22 ± 0.41 mm for Group 0 and 0.64 ± 0.29 mm for Group 1. Group 2 showed a creeping tendency of 0.77 ± 0.22 mm (Table 1, Figure 10), where creeping represents coronal growth of the gum margin without any sign of inflammation. Since some patients received more than an implant, the individual characteristics of the persons were tested and statistically did not significantly affect BBT (*p* = 0.35) nor ΔTH at three years (*p* = 0.56). Thirty-two implants were placed in the upper jaw and 46 in the lower. Positioning in the lower or upper dental arch did not affect the ΔTH at three years (*p* = 0.26). Positions ranging between the first premolars were considered as the anterior aesthetic area. Implants in the anterior zone were 14 and the fixtures in the posterior area were 64. The anterior/posterior positioning statistically affected BBT (*p* = 0.005) and ΔTH at three years (*p* = 0.008). Tissue remodeling showed greater values in the anterior zone, as detailed in Table 2.

While correlation between BBT and ΔTH at three years in the posterior zone was very significant (*p* ≤ 0.001), correlation in the anterior area could not be found (*p* = 0.21). It is mandatory to take into consideration that the small sample size of the anterior implants group dramatically reduces the chance to find any correlation. In addition, the average BBT in the anterior area was only 0.5 mm, which might explain the greater tendency for recession. According to the present findings, the null hypothesis was confirmed: a correlation was found between the BBT and the TH over three years; nevertheless, a wider sample is needed to assess this correlation in the anterior zone.

## 4. Discussion

Nowadays, aesthetic outcome plays a fundamental role in oral implantology [2]. Thus, current research has turned its attention to the parameters that determine the aesthetic success of rehabilitation [10]. Certainly, the height of the soft tissue around a dental implant is key in determining a natural aesthetic result [28], as no rehabilitation showing recessions can be considered an aesthetic achievement [28]. Accordingly, the amounts of hard and soft tissue around a dental implant are clearly very important [2]. However, the current literature does not indicate a clear relationship between those two variables. Buser et al. stressed the importance of a facial bone wall of sufficient height and thickness to obtain long-term stability of the gingival margins around implant restorations. If the buccal bone is missing and no reconstructive procedures are applied, the rehabilitation will frequently result in soft tissue recession, thereby compromising the aesthetics [14]. Buser et al. also recommended a minimum buccal bone thickness of 1 mm. However, their study defines “comfort” and “danger” zones for 3D implant positioning based on personal experience and a restorative and biology-driven philosophy rather than a scientific evaluation supported by clinical measurements. Similar to the observations of Buser et al., the present study results also indicated that the buccal bone thickness at the time of implant placement was critical for gum parabola stability over time. In particular, the current prospective data analysis revealed a strong correlation (*p* ≤ 0.001) between the BBT and the tendency of the buccal gum to reposition coronally or apically over a maturation time of three years. Spray et al. also investigated the influence of the BBT and bone quality on buccal bone vertical loss in a multicentric nonrandomized study of 2685 implants, where the vertical bone dimension was measured at the time of the initial surgery and compared with that measured at the second-stage surgery three to six months later [15]. They found a significant correlation between the BBT and the vertical buccal bone stability or loss: when the BBT was <1.8 mm, vertical buccal bone resorption was recorded, whereas BBT ≥1.8 mm showed more predictable bone stability at the second surgery. In the above-mentioned study, a caliper was used to measure the BBT after the final osteotomy yet prior to the implant placement. However, the tapping activity related to implant placement may inevitably reduce this measurement according to the apical thread design of the implant. As such, the actual BBT after implant placement may be reduced, making the considered values slightly overestimated. Moreover, this applies to both the vertical buccal bone height and the BBT, which are already known to affect the soft tissue response, as confirmed by Fabris et al. [29]. Thus, in the present study, the BBT was measured after implant insertion, and since the implants were positioned 0.5 mm subcrestal, the caliper was seated on the implant shoulder, resulting in true and repeatable measurements. More recently, Arora et al. conducted a two-year prospective study of 18 implants in the aesthetic zone [30]. However, they found no significant correlation between the BBT (range 0.45–1.24 mm) and soft tissue or aesthetic changes, referred to as the pink aesthetic score (PES) [1]. This lack of correlation may have been due to the low number of implants considered in their study and their use of immediate placement rather than implantation in healed sites. In their results, the mid-facial gingiva showed a mean recession of 0.06 ± 0.71 mm, so no statistical correlation could be found with the variability of such a small sample size. In the present study, to avoid the findings being affected by the depth of the implant positioning, only implants placed 0.5 mm below the crestal edge were included, in accordance with the manufacturer’s instructions. Notwithstanding, according to Siqueira et al., this consideration does not appear to be an influencing factor on the TH (*p* = 0.5) [31]. Another element affecting the current results may have been the implant connection [32]. In a systematic review of 29 articles, Goiato et al. revealed that internal connections induce lower marginal bone remodeling compared to external connections. Their results also indicated that the quality and quantity of bone affects the shape and maturation of the overlying soft tissues and, consequently, the pink aesthetics in the anterior region [32]. A study by Mangano et al. recommended morse-tapered connections for the anterior aesthetic zone based on stable peri-implant bone and aesthetic outcomes (pink aesthetic score and white aesthetic score evaluation at two years) for 26 anterior maxillary single-tooth implants [33]. Even though these above-mentioned studies used fresh socket positioning, it is still reasonable to assume that the implant connection will also influence tissue maturation in healed sites. All the implants included in the present study used the switching platform concept, which is already well-known to stabilize the peri-implant marginal bone, thereby enhancing the tissue response [34,35,36]. Another variable affecting the current results may have been the initial mucosa thickness [6,37]. Various articles have shown a significantly lower loss of crestal bone when the initial mucosal thickness surrounding a bone-level-positioned implant is ≥2 mm [38]. For this reason, all sites presenting less than 2 mm of crestal mucosa thickness were excluded from the measurements and statistical analysis. The surgical technique itself can similarly be an important factor stabilizing the tissue contours; for example, the use of a connective tissue graft [39,40]. Accordingly, the surgical technique used in the current study was deliberately simplified to eliminate additional variables that could have influenced the outcomes and to reduce the confounding factors. The prosthetic technique and materials used may also influence the soft and hard tissue remodeling. All implants were rehabilitated with a cement-retained metal-ceramic crown on milled abutments. While the crown never extends over 1 mm deep from the free gingival margin, the abutment holds a deeper position in the connective tissue and next to the bone. The abutment used in the present study is made of anodized titanium that becomes machined in the milled areas (Ti6Al4V, Titanium Grade 5). Brunello et al. investigated different characteristics of this material in a comparative analysis. Titanium Grade 5, both polished or anodized, resulted equally biocompatible to the proliferation and adhesion of human gingival fibroblasts and presented no hemolytic activity. It was also measured as an antibacterial activity of titanium to several pathogens, involved in peri-implant infections, in favor of the anodized treatment [41]. Another possible limitation of the present study is represented by the fact that the biotype assessment was not included in the statistical analysis. Kloukos, in a cross-sectional comparative study, indicates the transgingival probing with the periodontal probe an adequate choice for the tissue thickness determination [42]. While it seems solid the fact that bone remodeling might be associated with thin biotype, the association with the free gingival margin stability around dental implants is still weak. This aspect might be further investigated in the long term with a wide range of samples [43]. The majority of the implants considered in this study are located in the posterior area due to the consecutive enrolment. It seems that the position may affect ΔTH. Analyzing the sample, it appears that in the anterior zone a thinner BBT is found, which probably is the reason of a higher recession at ΔTH.In addition, in this area, the sample size is inappropriate for representative statistics; further investigations with a wider sample size in the aesthetic area is needed. In the present study, the buccal free gingival margin was found to mature coronally when the BBT was ≥1.5 mm (Group 2), as also described in a previous case report by Pereira et al. [44]. Another study also reported that the buccal tissue thickness has a significant influence on its own height relative to the implant connection in order to maintain consistent proportions [10]. Hence, it is reasonable to conclude that increasing the buccal tissue thickness will induce increased tissue height and better stability over time. Notwithstanding, extensive evidence on tissue behavior is still lacking, so more long-term studies are needed with larger samples and increased variables and mechanisms. Consequently, based on the results for the current sample and within the limits of the present study, when the BBT is less than 1.5 mm after implant placement, a GBR procedure should seriously be considered to reduce the risk of buccal gum recession. However, additional studies are still needed to define the optimal BBT and thereby enhance the current knowledge of peri-implant tissue stability in different surgical, prosthetic, and tissue biotype situations. To increase the BBT in healed edentulous ridges, different countermeasures can be considered: socket preservation techniques to reduce alveolar ridge remodeling after extraction [7,8,45], reducing the implant diameter, or placing the implant in a more palatal/lingual position. Yet, care is needed, as more palatal/lingual implant positioning can also result in bone dehiscence and difficult prosthetic adaptation [46]. Finally, narrow implants have shown acceptable survival rates in a retrospective study of 335 implants with a seven-year follow-up [47] and in narrow ridges [48].

## 5. Conclusions

This prospective clinical study revealed the importance of the buccal bone thickness at implant placement as a key influence on the stability of the facial gingival margin over a maturation time of three years following the delivery of the final prosthodontics. Within the limits of the current study, the results indicate a minimum BBT of 1.5 mm to avoid the risk of gingival margin recession, which is considered critical in the aesthetic zone. Moreover, the current data will be useful to determine when bone augmentation is needed and to reduce the risk of implant dehiscence over time [18]. The appropriate implant diameter and positioning should also be determined to satisfy this minimal requirement. Plus, the implant design itself can play a critical role in bone remodeling and gingival recession [49]. In the aesthetic zone, the number of implants considered in this study is small and should be implemented for future confirmation. Further studies are needed to investigate the impact of the coronal implant geometry on preserving the BBT and the stability of peri-implant tissue in the different areas of the mouth.

## Figures and Tables

**Figure 1 materials-13-00511-f001:**
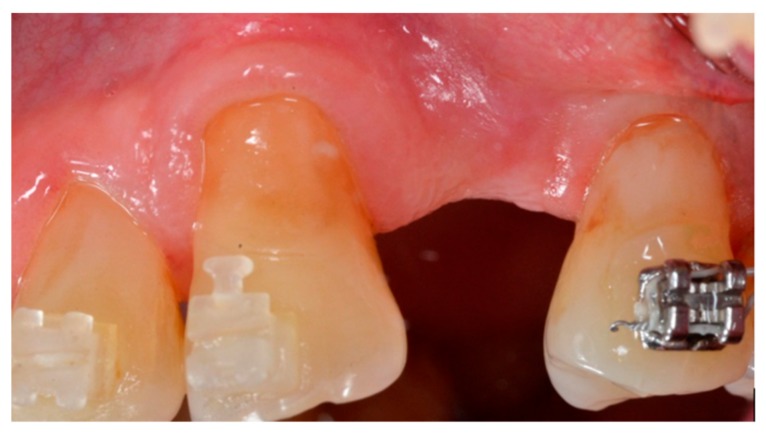
Clinical view.

**Figure 2 materials-13-00511-f002:**
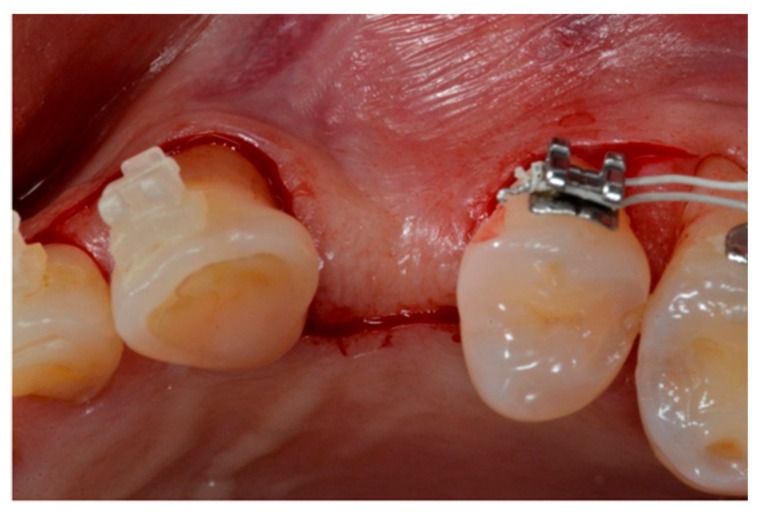
Flap incision: crestal access with intrasulcular and modified papilla preservation.

**Figure 3 materials-13-00511-f003:**
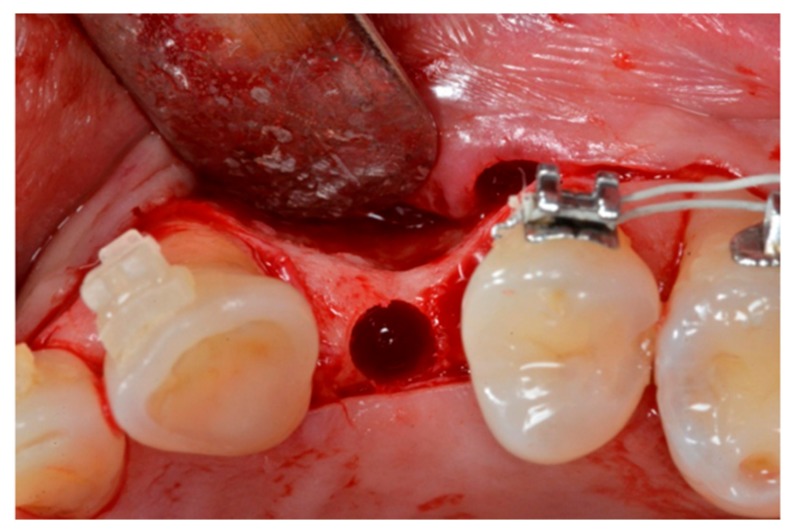
Prepared implant site.

**Figure 4 materials-13-00511-f004:**
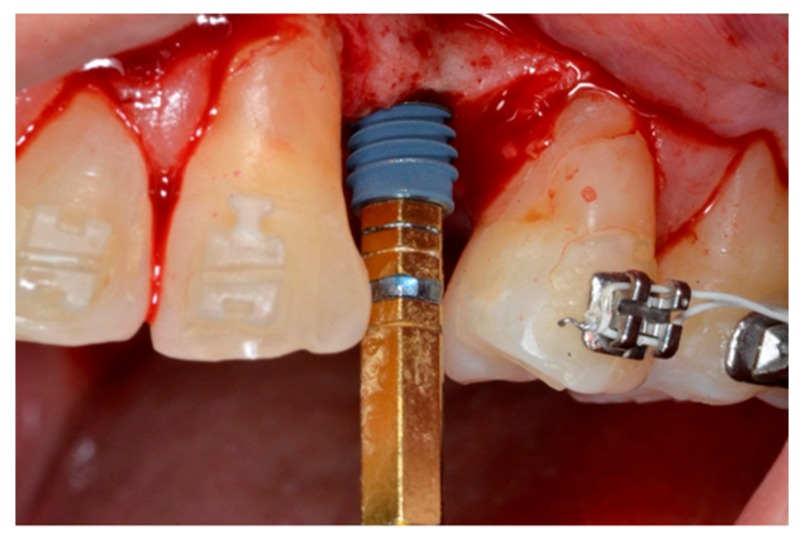
Implant placement.

**Figure 5 materials-13-00511-f005:**
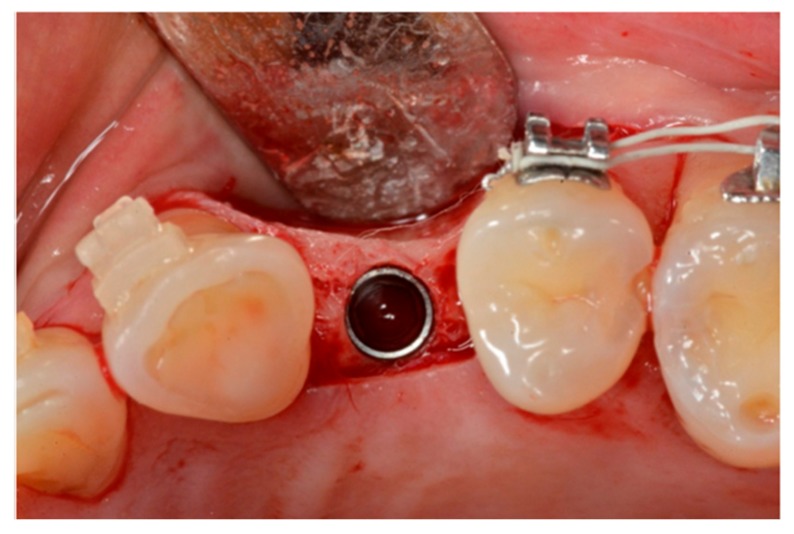
Implant positioning 0.5 mm under the bone edge.

**Figure 6 materials-13-00511-f006:**
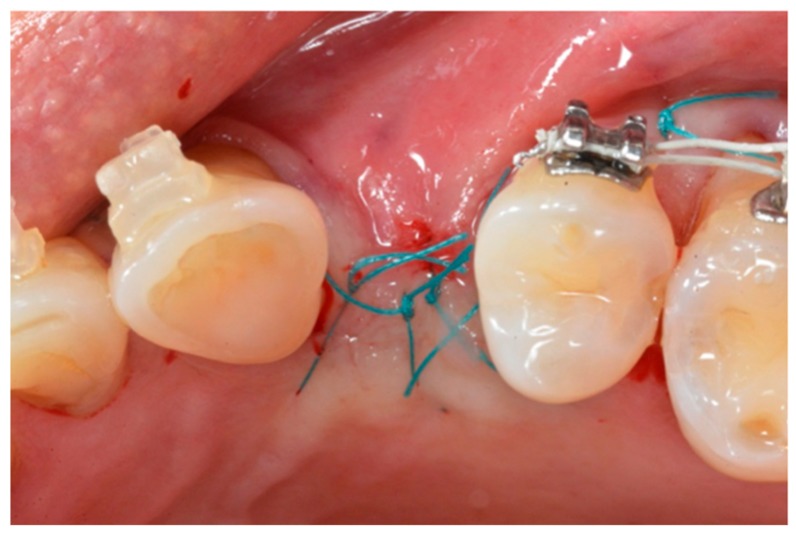
Free tension suture.

**Figure 7 materials-13-00511-f007:**
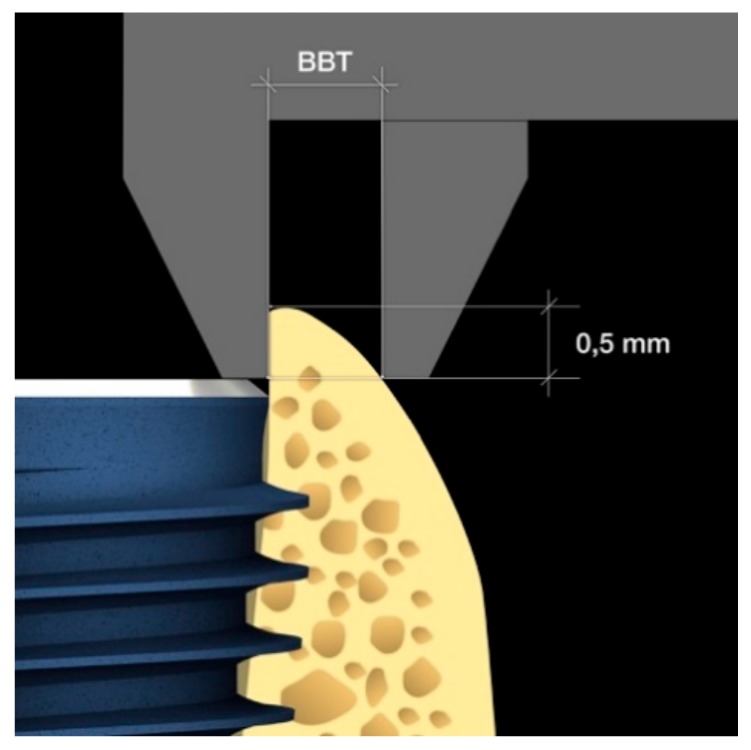
Graphic of buccal bone measurement above implant shoulder.

**Figure 8 materials-13-00511-f008:**
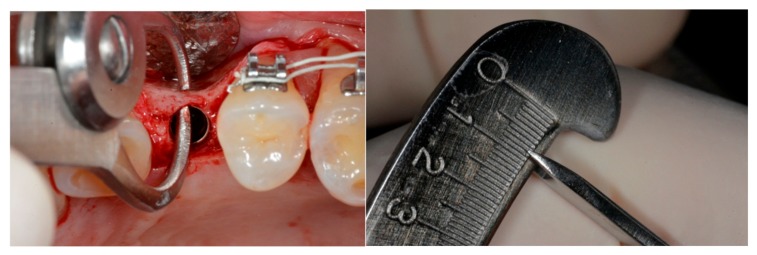
(**a**) Interoperative buccal bone thickness (BBT) measurement using a stainless steel dental gauge caliper and (**b**) a measurement value displayed on the gauge arc with 0.1 mm marks.

**Figure 9 materials-13-00511-f009:**
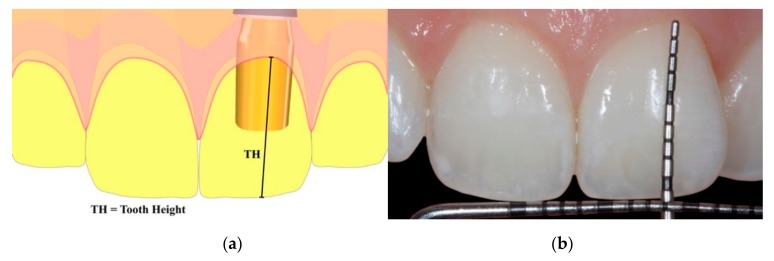
(**a**) Graphical representation of the measurement and (**b**) clinical view of the measurement. The tooth height (TH) is measured as the distance between the buccal gingival zenith and the crown incisal edge, according to the main axis of the tooth.

**Figure 10 materials-13-00511-f010:**
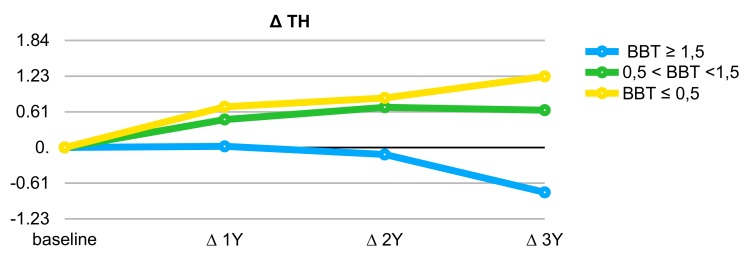
The average recession or creeping (∆TH) distribution pattern at one, two, and three years from the final crown placement according to the BBT group. Negative values correspond to a creeping attachment, positive values mean recession.

**Table 1 materials-13-00511-t001:** Average apical or coronal migration (creeping) of free gingival margin at buccal zenith level (∆TH) at one-, two-, and three-year follow-ups from delivery of final prosthodontics and relationship with BBT grouping. Negative values mean creeping. Statistical significance between tooth height (TH) variation and BBT grouping was found for all 78 implants at all follow-ups.

ΔTH Averages	∆TH 1Y	∆TH 2Y	∆TH 3Y
Group 2—BBT ≥1.5	−0.02 ± 0.45	−0.12 ± 0.71	−0.77 ± 1.12
Group 1—0.5 < BBT <1.5	0.48 ± 1.21	0.69 ± 1.58	0.64 ± 1.57
Group 0—BBT ≤ 0.5	0.7 ± 1.57	0.85 ± 1.98	1.22 ± 1.95
Number of implants	78	78	78
*p* (Pearson Two-Tailed)	0.032	0.025	*p* ≤ 0.001

**Table 2 materials-13-00511-t002:** Implant anterior/posterior distribution with average buccal bone thickness and deltas.

ΔTH Averages	Number of Implants	Average BBT	∆TH 1Y	∆TH 2Y	∆TH 3Y
Anterior	14	0.5	0.61	1.11	1.43
Posterior	64	1.2	0.33	0.33	0.10

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
