# Peer review of "Correlation between Buccal Bone Thickness at Implant Placement in Healed Sites and Buccal Soft Tissue Maturation Pattern: A Prospective Three-Year Study"

_materials, 2020, doi:10.3390/ma13030511_

Round 1

Reviewer 1 Report

This study investigated the association between facial/buccal bone thickness and gingival recession following implant placement and restoration. The patients were followed up for 3 years and the authors found that the facial or buccal bone thickness is associated with the stability of the gingival margin.

The English language and styles are horrendous and it needs extensive editing. There are many conjunctive adverbs such as therefore, nevertheless, still, moreover, however and many more, and they are often placed in very strange places.  Figure 2 was flipped 180 degrees and Figure 4 was rotated 90 degrees. They look odd. Table 1 is unnecessary. Buccal bone width, or BBW, is very confusing. It takes a while for me to realize that it actually refers to the thickness, not the width of the bone facial or buccal to the implants. Please change it to facial or buccal bone thickness. As this paper focuses on aesthetics, it is the anterior teeth that are truly important. Facial bone in the anterior is more relevant to esthetics than the buccal bone in the posterior part of the mouth. Anterior teeth, especially maxillary anterior teeth, should be identified and analyzed as a subgroup. Facial bone thickness should be separated from buccal bone thickness as they have entirely different significances as far as aesthetics is concerned. Gingival margin stability is not only associated with facial or buccal bone thickness. Soft tissue biotype probably plays a more important role. It is therefore pointless to discuss the effect of bone thickness without regard to soft tissue biotype. Is there any data collected on the tissue biotype?  Surgical techniques may affect the pattern of bone resorption after surgery. Figure 3-5 show that a full thickness flap with periosteal membrane was lifted and separated from the facial/buccal bone. This seems unnecessary as there are plenty of evidence that separation of periosteum from bone surface is detrimental to bone healing and may aggravate bone resorption. On page 5, line 170-171, it states that "all implants not showing a position within 0.5mm of the crest edge in the periapical radiograph were dropped out from the trial". But the result was not presented. How many cases were dropped out because of this criterium? The 78 implants were from 38 patients, which implies that in many cases there are more than one implant. This means that the implant as an analytical unit is not independent from the other implants. Please consult a statistician regarding statistical analyses of dependent data.

Author Response

Dear Editor and Reviewers,

            thank you for giving us the great opportunity to revise and resubmit for publication the revised version of the manuscript “materials- 694788”. In keeping with our last communication with you, we are resubmitting this revision according to the Journal guidelines before the agreed upon deadline (16/01/2020). Enclosed You will find the responses to the comments of the reviewers.

We are very glad you found interest in our clinical study and honestly we feel privileged to review the present manuscript.

Answers to Reviewer 1:

Kind Reviewer, thank you for the precious comments and guidance. Please find the answers point by point and the text updated accordingly in the paper.

Comment 1 - The English language and styles are horrendous and it needs extensive editing.

Answer  1 - The text has been corrected by a native English speaker friendly with the field and edited for improving the English language;

Comment 2 - There are many conjunctive adverbs such as therefore, nevertheless, still, moreover, however and many more, and they are often placed in very strange places.

Answer 2 – The number of adverbs throughout the text has been reduced and their place in the sentences reviewed;

Comment 3 - Figure 2 was flipped 180 degrees and Figure 4 was rotated 90 degrees. They look odd.

Answer 3 - Figure positioning has been improved and rotation corrected;

Comment 4 – Table 1 is unnecessary.

Answer 4 – Table 1 was eliminated, former table 2 became table 1. A new table regarding anterior / posterior positioning has been added and it’s now called table 2;

Comment 5 – Buccal bone width, or BBW, is very confusing. It takes a while for me to realize that it actually refers to the thickness, not the width of the bone facial or buccal to the implants. Please change it to facial or buccal bone thickness.

Answer 5 – Buccal bone width (BBW) has been changed with Buccal Bone Thickness (BBT) in the whole text and in the title of the paper, to improve the readability of the article;

Comment 6 – As this paper focuses on aesthetics, it is the anterior teeth that are truly important. Facial bone in the anterior is more relevant to esthetics than the buccal bone in the posterior part of the mouth. Anterior teeth, especially maxillary anterior teeth, should be identified and analyzed as a subgroup. Facial bone thickness should be separated from buccal bone thickness as they have entirely different significances as far as aesthetics is concerned.

Answer 6 – Major changes have been inserted in the result section: maxillary anterior teeth have been analyzed as a subgroup and the results have been inserted in the paper along with the statistical analysis of the group (lines 216-227, table 2, lines 304-309);

Comment 7 – Gingival margin stability is not only associated with facial or buccal bone thickness. Soft tissue biotype probably plays a more important role. It is therefore pointless to discuss the effect of bone thickness without regard to soft tissue biotype. Is there any data collected on the tissue biotype?

Answer 7 – Text has been edited, according to Linkevicious, van Eekeren, Klouklos findings (respectively: lines 115-116, ref [21], lines 281-283, ref n.38; lines 299-300, ref [42]), every patient who presented less than 2 mm of crestal mucosa thickness was considered as having a thin biotype and therefore excluded from the study;

Comment 8 – Surgical techniques may affect the pattern of bone resorption after surgery. Figure 3-5 show that a full thickness flap with periosteal membrane was lifted and separated from the facial/buccal bone. This seems unnecessary as there are plenty of evidence that separation of periosteum from bone surface is detrimental to bone healing and may aggravate bone resorption.

Answer 8 - Ideally that would be the best option, unfortunately, to precisely monitor the BBT and correctly position the implant, an as-minimal-as-possible full thickness flap has been elevated. The text has been updated to motivate the choice (lines 117-119);

Comment 9 – On page 5, line 170-171, it states that "all implants not showing a position within 0.5mm of the crest edge in the periapical radiograph were dropped out from the trial". But the result was not presented. How many cases were dropped out because of this criterium?

Answer 9 – This was a mistake in the writing, every implant which could not be placed according to the surgical protocol (e.g. excessive bone density, steep interproximal bone peak) was excluded and not counted in the statistical analysis (lines 85-88);

Comment 10 – The 78 implants were from 38 patients, which implies that in many cases there are more than one implant. This means that the implant as an analytical unit is not independent from the other implants. Please consult a statistician regarding statistical analyses of dependent data.

Answer 10 - individual characteristics of the persons have been tested and it was seen that, statistically, they did not affect significantly nor BBT (p= 0.35) nor ΔTH at three years (p=0.56), text had been updated accordingly (lines 212-214).

We are very glad you found interest in our clinical study and honestly we feel privileged to review the present manuscript.

Reviewer 2 Report

Dear authors,

The paper is interest for the readers, but to make a correct review, the authors need to send new paper because tables 1 and 2, page 7, are not possible to analise due the sobreposition of the tables and legends.

Author Response

Dear Editor and Reviewers,

            thank you for giving us the great opportunity to revise and resubmit for publication the revised version of the manuscript “materials- 694788”. In keeping with our last communication with you, we are resubmitting this revision according to the Journal guidelines before the agreed upon deadline (16/01/2020). Enclosed You will find the responses to the comments of the reviewers.

Answers to Reviewer 2:

Kind Reviewer, thank you for the precious comments and guidance. Please find the answers point by point and the text updated accordingly in the paper.

The paper is interest for the readers, but:

Comment 1 - to make a correct review, the authors need to send new paper because tables 1 and 2, page 7, are not possible to analise due the sobreposition of the tables and legends.

Answer 1 – text, figures and tables have been thoroughly edited, now it should be possible to correctly view them.

We are very glad you found interest in our clinical study and honestly we feel privileged to review the present manuscript.

Reviewer 3 Report

Dear Authors,

 After the review process, I have several comments: you should move the last phrase from introduction (Page 2, Line 60), because it is a conclusion/perspectives of the study; you should correct the explanation of Figure 2; you should insert references in all sections from Materials and Methods; you should improve figures 8 and 9 for a better understanding of the explanations; you should declare if the figures 7 and 8 are original or adapted from other sources; you should reorganize Results section, because it is hard to read it; you should insert more new references, because part of them are too old (before 2010); you should detail the correlation between structural characteristic and biological effects (e.g., possible antimicrobial effects); you should insert more comments about the stability of the material/product and possible toxicity.

Best regards!

Author Response

Dear Editor and Reviewers,

            thank you for giving us the great opportunity to revise and resubmit for publication the revised version of the manuscript “materials- 694788”. In keeping with our last communication with you, we are resubmitting this revision according to the Journal guidelines before the agreed upon deadline (16/01/2020). Enclosed You will find the responses to the comments of the reviewers.

Answers to Reviewer 3:

Kind Reviewer, thank you for the precious comments and guidance. Please find the answers point by point and the text updated accordingly in the paper.

Comment 1 - you should move the last phrase from introduction (Page 2, Line 60), because it is a conclusion / perspectives of the study;

Answer 1 – the sentence has been moved to the conclusions and rewritten (line 331-332)

Comment 2 - you should correct the explanation of Figure 2.

Answer 2 - The text has been updated explaining the surgical design of the flap

Comment 3 - you should insert references in all sections from Materials and Methods.

Answer 3 – new references have been added in the text (line 52, ref [16] line 97, ref [19], line 99, ref [20], line 120, ref [22], line 281, ref [37], line 293, ref [41], line 299, ref. [42])

Comment 4 - you should improve figures 8 and 9 for a better understanding of the explanations.

Answer 4 - text has been improved for a better understanding of the figures

Comment 5 - you should declare if the figures 7 and 8 are original or adapted from other sources

Answer 5 – all pictures, figures and tables are originally captured/drawn by dr. Davide Farronato. Author contributions section has been updated with this specification;

Comment 6 - you should reorganize Results section, because it is hard to read it

Answer 6 – The results section has been edited; the text should now be more clearly understandable;

Comment 7 - you should insert more new references, because part of them are too old (before 2010);

Answer 7 – References older than 2010 were substituted with more recent ones. Some old references were not changed since they were considered milestones and, in the authors opinion, needed to be cited (e.g. Araujo, line 319-321, ref [45]);

Comment 8 - you should detail the correlation between structural characteristic and biological effects (e.g., possible antimicrobial effects);

Answer 8 – the biological relationship and antimicrobial effects had been disclosed in the text (lines 291-297, ref [41]);

Comment 9 - you should insert more comments about the stability of the material/product and possible toxicity.

Answer 9 –  the biocompatiblity of the material, the proliferation and adhesion of human gingival fibroblasts has been updated. (lines 291-297, ref [41], lines 97-100, ref [19] [20]).

We have made every effort to accommodate the recommendations of the Editor and Reviewers. Therefore, we hope that this revision will move this manuscript closer to publication in your prestigious Journal. Again, thank you for your precious consideration and interest in the present paper and please do not hesitate to contact us if you have any further questions or concerns.

Sincerely,

Mattia Manfredini

Round 2

Reviewer 2 Report

Dear authors,

Congratulations and thank you for your submission to MDPI journal. The work is significant and with interest for the readers.

Reviewer 3 Report

Dear Editor,

I do not have any supplementary comments.

Best regards!